# Early preclinical detection of prions in the skin of prion-infected animals

Zerui Wang[1,2], Matteo Manca[3], Aaron Foutz[4], Manuel V. Camacho[1], Gregory J. Raymond[3],
Brent Race[3], Christina D. Orru[3], Jue Yuan[1], Pingping Shen[1,2], Baiya Li[1,5], Yue Lang[1,2], Johnny Dang[1],
Alise Adornato[1], Katie Williams[3], Nicholas R. Maurer[1], Pierluigi Gambetti[1], Bin Xu[6], Witold Surewicz[7],
Robert B. Petersen[1,8], Xiaoping Dong[9], Brian S. Appleby[1,4,10], Byron Caughey [3], Li Cui[2],
Qingzhong Kong[1,4,10,11] & Wen-Quan Zou[1,2,4,9,10,11]

A definitive pre-mortem diagnosis of prion disease depends on brain biopsy for prion detection currently and no validated alternative preclinical diagnostic tests have been reported to date. To determine the feasibility of using skin for preclinical diagnosis, here we report ultrasensitive serial protein misfolding cyclic amplification (sPMCA) and real-time quaking-induced conversion (RT-QuIC) assays of skin samples from hamsters and humanized transgenic mice (Tg40h) at different time points after intracerebral inoculation with 263K and sCJDMM1 prions, respectively. sPMCA detects skin PrP$^{Sc}$ as early as 2 weeks post inoculation (wpi) in hamsters and 4 wpi in Tg40h mice; RT-QuIC assay reveals earliest skin prion-seeding activity at 3 wpi in hamsters and 20 wpi in Tg40h mice. Unlike 263K-inoculated animals, mock-inoculated animals show detectable skin/brain PrP$^{Sc}$ only after long cohabitation periods with scrapie-infected animals. Our study provides the proof-of-concept evidence that skin prions could be a biomarker for preclinical diagnosis of prion disease.

[1] Department of Pathology, Case Western Reserve University School of Medicine, Cleveland 44106 OH, USA. [2] Department of Neurology, The First Hospital of Jilin University, Changchun 130021 Jilin Province, The People's Republic of China. [3] Laboratory of Persistent Viral Diseases, Rocky Mountain Laboratories, National Institute of Allergy and Infectious Diseases (NIAID), National Institutes of Health (NIH), Hamilton 59840 MT, USA. [4] National Prion Disease Pathology Surveillance Center, Case Western Reserve University School of Medicine, Cleveland 44106 OH, USA. [5] Department of Otolaryngology, The First Affiliated Hospital of Xi'an Jiaotong University, Xi'an 710061 Shanxi Province, The People's Republic of China. [6] Department of Biochemistry, Virginia Polytechnic Institute and State University, Blacksburg 24061 Virginia, USA. [7] Department of Physiology and Biophysics, Case Western Reserve University School of Medicine, Cleveland 44106 OH, USA. [8] Foundational Sciences, Central Michigan University College of Medicine, Mount Pleasant 48859 MI, USA. [9] State Key Laboratory for Infectious Disease Prevention and Control, National Institute for Viral Disease Control and Prevention, Chinese Center for Disease Control and Prevention, Beijing 102206, The People's Republic of China. [10] Department of Neurology, University Hospitals Cleveland Medical Center, Case Western Reserve University School of Medicine, Cleveland 44106 OH, USA. [11] National Center for Regenerative Medicine, Case Western Reserve University School of Medicine, Cleveland 44106 OH, USA. These authors contributed equally: Zerui Wang, Matteo Manca, Aaron Foutz, Manuel V. Camacho. Correspondence and requests for materials should be addressed to B.C. (email: bcaughey@nih.gov) or to L.C. (email: chuili1967@126.com) or to Q.K. (email: qxk2@case.edu) or to W.-Q.Z. (email: wxz6@case.edu)

Prion diseases are fatal transmissible neurodegenerative diseases affecting both humans and animals. They include scrapie in sheep and goats, bovine spongiform encephalopathy (BSE) in cattle, chronic wasting disease (CWD) in elk, deer, and moose, as well as Creutzfeldt-Jakob disease (CJD), kuru, fatal insomnia, Gerstmann–Sträussler–Scheinker syndrome and variably protease-sensitive prionopathy in humans[1]. All these diseases are associated with the infectious misfolded form (prion or PrP$^{Sc}$) of a cellular prion protein (PrP$^C$) through a high α-helix structure to high β-sheet conformational transition[2,3].

Although PrP$^C$ is expressed in virtually all tissues and organs, PrP$^{Sc}$ is mainly deposited in the central nervous system and linked to the spongiform degeneration and neuronal loss that are the neuropathological hallmarks of prion diseases. A definitive diagnosis of prion disease has historically mainly depended on examination of brain tissues obtained by biopsy or at autopsy for the presence of prions and neuropathological changes. The recent development of cerebrospinal fluid (CSF)- and nasal brushings-based real-time quaking-induced conversion (RT-QuIC) analysis provides an alternative approach[4–6], but to date has only been validated for the diagnosis of clinical-stage prion disease[4–9]. Moreover, lumbar puncture for CSF sampling is not feasible for all patients due to contraindications and complications in certain conditions. RT-QuIC and serial protein misfolding cyclic amplification (sPMCA) analysis of urine and blood have not been helpful for diagnosing sCJD patients. For instance, PrP$^{Sc}$ was detectable in the urine from patients with variant CJD (vCJD, a distinct CJD that originated from exposure to BSE), but, was at much lower rate in sCJD[10]. Using blood-based sPMCA, PrP$^{Sc}$ was detectable in vCJD and in only 1 out of 67 sCJD patients[11], consistent with the finding that prion transmission between individuals through blood transfusion has only been reported for vCJD but not sCJD[12]. There is a need to seek additional readily accessible specimens for the detection of prions in preclinical diagnosis, and for monitoring of disease progression and therapeutic efficiency.

Our recent finding of infectious prions in the skin of patients with sCJD and vCJD raised the possibility that skin PrP$^{Sc}$ could be used as a biomarker for early diagnosis and assessment of disease progression[13]. To test this hypothesis, we examine skin PrP$^{Sc}$ in hamsters and humanized transgenic (Tg) mice at different time points after intracerebral prion inoculation using the highly sensitive sPMCA and RT-QuIC assays. We reveal that PrP$^{Sc}$ can be detected in the skin of scrapie-infected hamsters at 2 weeks post inoculation (wpi) and human prion-infected humanized Tg mice at 4 wpi by sPMCA as well as skin prion-seeding activity be detected at 3 wpi in hamsters and 20 wpi in Tg mice by RT-QuIC assay. It is worth noting that, compared to the 263K-inoculated hamsters, the mock-inoculated hamsters that has a longer cohabitation period with infected animals exhibit amplified PrP$^{Sc}$ by sPMCA in both skin and brain tissues without clinical signs or detectable brain PrP$^{Sc}$ by conventional western blotting.

## Results

### Detection of skin PrP$^{Sc}$ by sPMCA in prion-infected hamsters.
The inoculated animals were killed at 0.4, 1, 2, 3, 4, 7, 10, and 11 weeks post inoculation (wpi) with the 263K scrapie strain, and skin and brain tissues were collected. Clinical signs of scrapie-infected animals appeared from 10 wpi. The deposition of PrP$^{Sc}$ and spongiform degeneration became detectable by conventional western blotting, immunohistochemistry, and H&E staining in the brain at 4 wpi and 7 wpi, respectively (Supplementary Figs. 1, 2).

The skin samples from thigh, back, and belly areas of 263K-inoculated hamsters were examined using serial protein misfolding cyclic amplification (sPMCA), a highly sensitive assay for

amplification of small amounts of PrP$^{Sc}$[14,15]. In the thigh skin of scrapie-inoculated animals, the PrP$^{Sc}$ was first detected by western blot in the 4th round of sPMCA for 2 wpi samples, in the 3rd round of sPMCA for 7 wpi samples, and in the 2nd round of sPMCA for 10 and 11 wpi samples (Fig. 1a, b, Supplementary Fig. 3), indicating increasing levels of PrP$^{Sc}$ over time. The same trend was observed in skin samples from the back and the belly areas of infected animals (Fig. 1b). In contrast, no PrP$^{Sc}$ was detected after 8 rounds of sPMCA in the skin of thigh, back and belly from 0.4 to 1 wpi 263K-inoculated hamsters (Supplementary Fig. 4) or 12 wpi-negative control animals intracerebrally inoculated with PBS (Supplementary Fig. 5). Therefore, PrP$^{Sc}$ became detectable by sPMCA in the skin of the scrapie-infected animals 5 weeks ahead of brain spongiosis and 8 weeks before the first signs of clinical symptoms.

### Detection of skin PrP$^{Sc}$ by RT-QuIC in infected hamsters.
Skin samples from thigh, back, and belly of 263K-inoculated hamsters were also subjected to RT-QuIC assays using recombinant hamster PrP90-231 as the substrate[13]. Prion-seeding activity was detected in skin samples from the back starting from as early as 3 wpi, but it was not detectable in skin samples from the thigh and belly areas until near terminal stages (Fig. 2).

In a parallel RT-QuIC study, an independent set of hamsters was inoculated and examined with RT-QuIC by a second laboratory (the Caughey laboratory) where the RT-QuIC technology was initially co-developed[4,16]. In addition to the brain and thigh skin samples, skin tissues from ear pinna, and lateral rib cage area were also collected and analyzed. The earliest detection of prion-seeding activity was found in one of four ear pinna samples collected at 3 wpi (Fig. 3). Earliest detection in the skin collected from the rib cage area was achieved at 7 wpi while unambiguous RT-QuIC signals were seen starting from 9 wpi in the thigh samples (Fig. 3). End-point dilution RT-QuIC reactions of 11–12 wpi tissues indicated that the average prion-seeding activity in skin samples was about $10^3$- to $10^5$-fold lower than that in brain tissues, detected by two-independent laboratories (Tables 1, 2).

### Detection of skin PrP$^{Sc}$ by sPMCA and RT-QuIC in Tg mice.
To determine whether skin PrP$^{Sc}$ has diagnostic potential for human prion patients, we examined sporadic CJD-inoculated humanized Tg40h mice that express normal human PrP with methionine at the polymorphic residue 129 of PRNP[17]. The Tg40h mice were inoculated with brain homogenate from a patient with sCJDMM1 [the most common human prion subtype with type 1 PrP$^{Sc}$ and methionine/methionine at PrP residue 129 of PRNP]. The infected mice showed clinical signs at 24 wpi, but brain PrP$^{Sc}$ and spongiform degeneration were detectable at 20 wpi by conventional western blotting and neurohistology (Supplementary Figs. 6, 7).

PrP$^{Sc}$ in the skin of sCJDMM1-infected Tg40h mice was examined with sPMCA and RT-QuIC assays. In the belly skin, PrP$^{Sc}$ was first detected by western blotting in the 4th round of sPMCA for 24 wpi samples and in the seventh round of sPMCA for 4-wpi samples (Fig. 4a, Supplementary Fig. 8). In contrast, no PrP$^{Sc}$ was detected in the belly skin of PBS-inoculated mice after eight rounds of sPMCA (Fig. 4b, Supplementary Fig. 8). In the thigh skin, PrP$^{Sc}$ was detected after seven or eight rounds of sPMCA for 4-wpi samples from sCJDMM1-inoculated mice, but not in skin samples from PBS-inoculated control animals (Fig. 4c, Supplementary Fig. 8).

No prion-seeding activity was detected by RT-QuIC with the SHaPrP90-231 as the substrate in skin samples from sCJDMM1-inoculated mice until 20 wpi (Fig. 5a–d). To determine whether

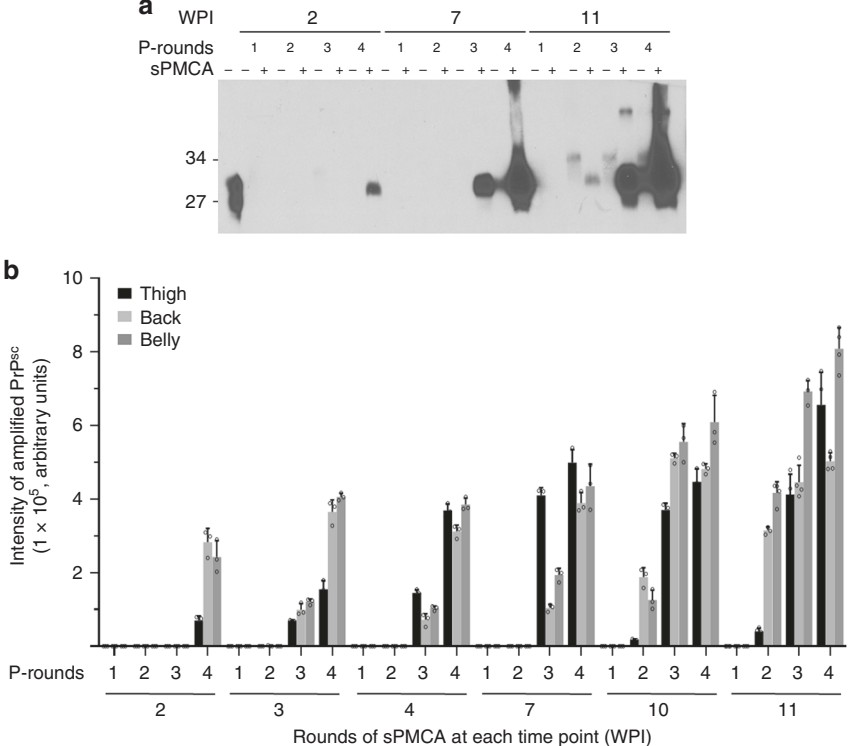

**Fig. 1** Western blot analysis of skin PrP$^{Sc}$ amplified by sPMCA. **a** Representative western blot analysis of amplified PrP$^{Sc}$ from the thigh skin of hamsters killed at 2, 7, and 11 weeks post inoculation (wpi) of 263K prions using four rounds of sPMCA. P-rounds: number of sPMCA rounds. All samples were treated with PK at 100 μg/ml except for the sample in the first lane of the blot that was used as a non-PK treatment control. The blot was probed with 3F4. The molecular weight markers are shown in kDa on the left side of the blot. **b** Quantitative analysis of total western blot results on the sPMCA-amplified PrP$^{Sc}$ from the thigh, back, and abdominal skin tissues of infected hamsters at 2 ($n = 3$), 3 ($n = 3$), 4 ($n = 3$), 7 ($n = 3$), 10 ($n = 3$), and 11 ($n = 4$) wpi. Data are presented as mean ± s.d.; the error bars represent the standard deviations of the intensity of amplified PrP$^{Sc}$ bands measured independently at each sPMCA round, which were from three-independent experiments

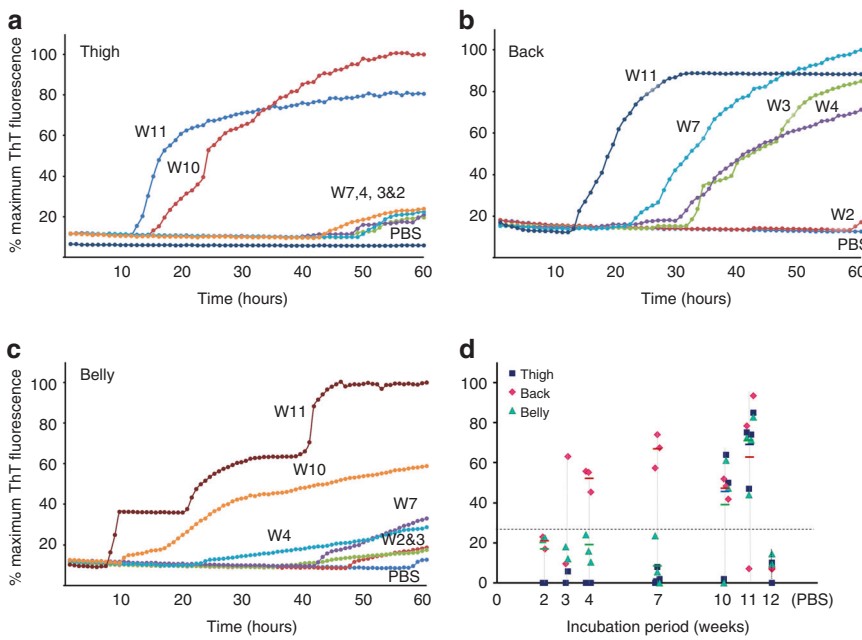

**Fig. 2** Skin prion-seeding activity of infected hamsters at different time points. Prion-seeding activity of thigh **a**, back **b**, and belly **c** skin from 263K-inoculated hamsters sacrificed at 2, 3, 4, 7, 10, or 11 wpi and PBS-inoculated hamsters at 12 wpi detected by RT-QuIC assay. **d** Scatter plot of skin prion-seeding activity of thigh, back, or belly from 263K-inoculated hamsters at 2 ($n = 2$), 3 ($n = 2$), 4 ($n = 3$), 7 ($n = 3$), 10 ($n = 3$), or 11 ($n = 4$) wpi and PBS-inoculated hamsters killed at 12 ($n = 2$) wpi. The horizontal dotted line indicates the 27% calculated ThT fluorescence threshold based on the means of negative controls plus 4 s.d.

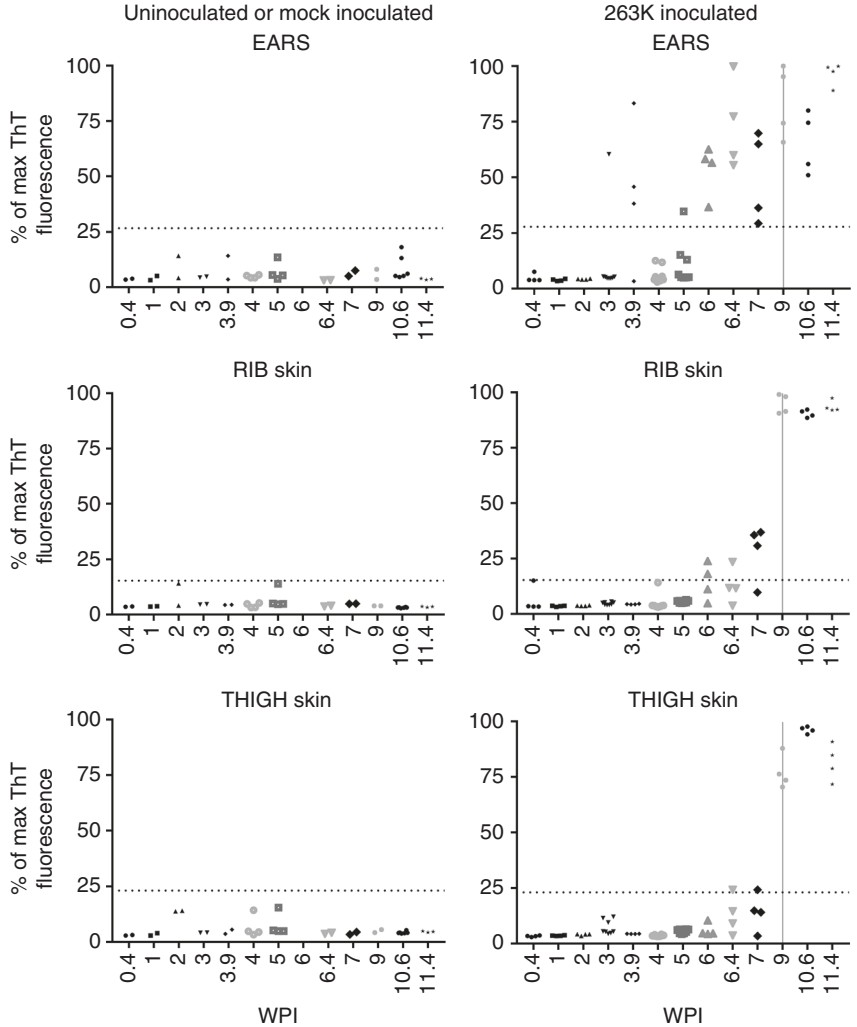

**Fig. 3** RT-QuIC analysis of ear pinna as well as rib and thigh skin. Left panel graphs show the results from either un-inoculated or mock-inoculated animals killed at the designated WPI while right panels show the results from inoculated animals. Each marker is an individual sample and represents the averaged ThT fluorescence from 8 replicate wells. Horizontal dotted line represents positivity threshold calculated as described in Materials and Methods. Vertical gray line indicates the time of appearance of clinical signs. All samples were tested in a blinded fashion

| Table 1 Comparison of prion-seeding activity between brain and skin of infected hamsters (LogSD$_{50}$/mg tissue)* | | |
|---|---|---|
| | **Brain (FC)** | **Skin (Belly)** |
| Hamster (12 wpi) | 10.50 | 6.05 |
| | 9.31 | 5.81 |
| | 9.31 | 5.31 |
| Tg40h (24 wpi) | 8.50 | 4.50 |
| | 7.75 | 3.45 |
| | 7.45 | 3.40 |
| *wpi* weeks post inoculation | | |
| *Conducted in the Zou laboratory | | |

| Table 2 Comparison of prion-seeding activity between brain and skin of infected hamsters (LogSD$_{50}$/mg tissue)* | | | |
|---|---|---|---|
| **Brain** | **Ear** | **Rib** | **Thigh** |
| 8.75 | 5.00 | 3.75 | 3.75 |
| 9.25 | 5.75 | 4.50 | 5.50 |
| 8.75 | 4.25 | 4.75 | 4.50 |
| 9.50 | 4.25 | 3.75 | 5.25 |
| *Conducted in the Caughey laboratory with hamsters at 11 weeks post inoculation | | | |

the use of recombinant human PrP as the substrate could increase sensitivity, we examined the back skin samples from infected Tg40h mice at different time points with the HuPrP(90–231)-based RT-QuIC assay. Similar to results shown in Fig. 5b, prion-seeding activity was detected in the back skin of Tg40h mice at 20 wpi and afterwards, but not in the skin of mice at 16 wpi or earlier (Fig. 5e).

The prion-seeding activities in the belly skin and brain samples from the sCJDMM1-inoculated Tg40h mice were quantified and

compared with RT-QuIC end-point titration assay. As shown in Fig. 6 and Table 1, relative to the brain concentrations of $10^9$–$10^{10}$ SD$_{50}$ per milligram of tissue (Fig. 6a), the belly skin samples contained $10^3$–$10^4$ SD$_{50}$ per milligram of tissue equivalent (Fig. 6b). Therefore, the average prion-seeding activity in the belly skin of infected Tg40h mice was ~$10^3$–$10^5$-fold lower than in the corresponding brain tissue samples (Table 1).

**Skin PrP$^{Sc}$ in controls co-housed with prion-infected animals.** Feces and urine from prion-affected hamsters and other animals

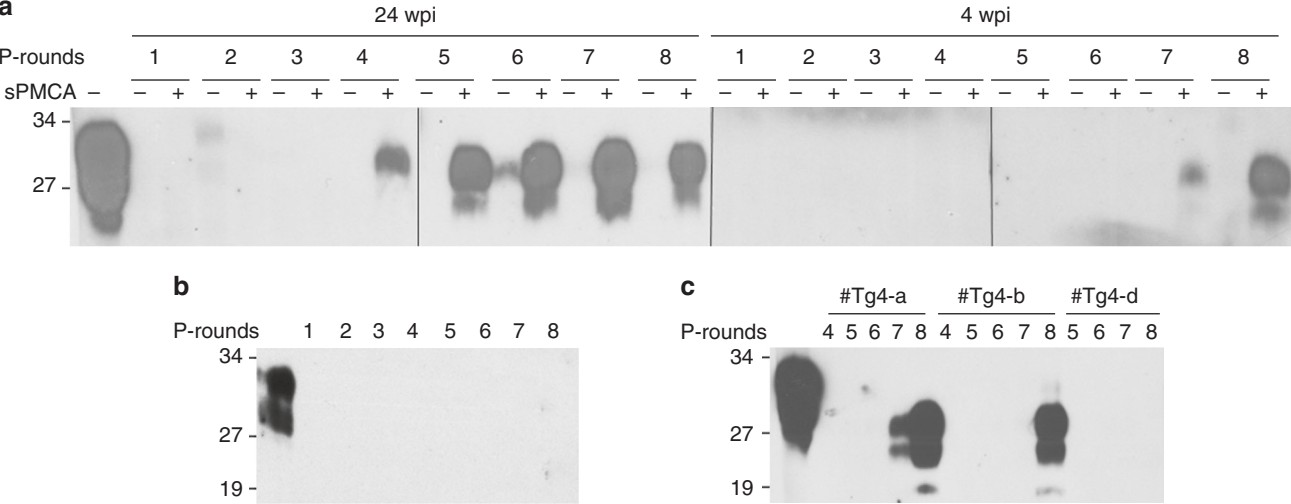

**Fig. 4** Detection of skin PrP$^{Sc}$ by sPMCA in the Tg40h mice. **a** Representative western blotting of PrP$^{Sc}$ amplified by eight rounds of sPMCA with belly skin homogenate from humanized Tg40h mice inoculated with sCJDMM1 homogenate and killed at 28 wpi or 4 wpi. **b** Representative western blotting of the belly skin samples from PBS-inoculated Tg40h mice after 8 rounds of sPMCA from three-independent experiments. **c** Representative western blot analysis of amplified PrP$^{Sc}$ from the thigh skin of prion-infected Tg40h mice killed at 4 wpi (#Tg4-a or #Tg4-b) by 4–8 rounds of sPMCA from three-independent experiments. #Tg4-d was the PBS-inoculated Tg40h mouse control in cohabitation with sCJDMM1-inoculated mice. P-rounds: number of sPMCA rounds. All samples were treated with PK at 100 µg/ml except for the sample in the first lane of the blots that was used as a non-PK treatment control. The blots were probed with 3F4. The molecular weight markers are shown in kDa on the left side of the blots

have been shown to contain PrP$^{Sc}$ and infectivity[18–24]. Thus, it is possible that the PrP$^{Sc}$ or prion-seeding activities that we detected by sPMCA or RT-QuIC in the skin of prion-inoculated animals could simply be due to external contamination of prions dropped/shed in the cages by the inoculated animals rather than true infection of the skin tissue. To address this concern, we examined the brain and skin tissues for PrP$^{Sc}$ from the PBS-inoculated hamsters that had been continuously co-habitated with the 263K-inoculated hamsters. No PK-resistant PrP$^{Sc}$ was detected in the brain of all PBS-inoculated cohabitating control hamsters by conventional western blotting until 12 wpi when they were all sacrificed for collection of tissues (Supplementary Fig. 1, lanes #wpi2-d, #wpi3-d, #wpi4-d, #wpi7-d, #wpi10-d, and #wpi11-d, Supplementary Fig. 2). In contrast, the PK-resistant PrP$^{Sc}$ was detected in the brain of scrapie-inoculated animals as early as 4 wpi (Supplementary Fig. 1, lanes #wpi4-b, #wpi4-c, Supplementary Fig. 2).

In the brain samples from cohabitating-negative controls, while PrP$^{Sc}$ was not amplified by sPMCA at 4 wpi or earlier (Fig. 7a, Supplementary Fig. 9), it was detected after 8 rounds of sPMCA for 7 wpi samples and after 3 rounds of sPMCA for 10 wpi samples (Fig. 7b, c, Supplementary Fig. 9). Remarkably, PrP$^{Sc}$ or prion-seeding activity was also amplified or detected in the back skin of cohabitating PBS-inoculated control hamsters after 3 rounds of sPMCA or by RT-QuIC at the terminal stage (11 wpi) but not at 10 wpi or earlier (Fig. 7d, Supplementary Fig. 9, Fig. 7e). In contrast, no PrP$^{Sc}$ was detected in brain or skin samples at any time point in PBS-inoculated control hamsters housed separately [Supplementary Fig. 1, lanes #wpi12 (PBS)-a, #wpi12 (PBS)-b, Supplementary Fig. 2, Supplementary Fig. 5].

## Discussion

Several lines of evidence have recently suggested that skin is the place where misfolded proteins often stay, which may play a role in the pathogenesis and early detection of neurodegenerative diseases[25]. While it has been known for a long time that sheep and goats with scrapie often have skin lesions[26], prions had not been detected in skin until prion infectivity was first found in skin

of prion-infected greater kudu using an animal-based bioassay[27]. Subsequently, skin PrP$^{Sc}$ was detected directly by western blotting after the enrichment of PrP$^{Sc}$ in experimentally or naturally scrapie-infected hamsters and sheep, as well as in a single cadaver with vCJD[28,29]. We recently observed both prion-seeding activity and prion infectivity in the skin of patients with sCJD and vCJD at the terminal stage of the diseases using RT-QuIC assay and a bioassay humanized Tg mouse-based, respectively[13]. In the current study, we further demonstrated that skin PrP$^{Sc}$ is pre-clinically detectable not only by RT-QuIC, but also by sPMCA before brain damage occurs in two animal models of prion diseases, 263K-inoculated hamsters and sCJDMM1-inoculated humanized Tg40h mice, with parallel RT-QuIC findings in an independent set of scrapie-infected hamsters done in an independent laboratory.

In terms of the earliest time point at which skin PrP$^{Sc}$ becomes detectable in animals infected by the intracerebral inoculation of prions, sPMCA showed detection at 2 wpi for hamsters and 4 wpi for Tg40h mice, while RT-QuIC detection was at 3 wpi for hamsters and 20 wpi for Tg40h (Fig. 8a). These findings indicate that skin PrP$^{Sc}$ is detectable at least 5 weeks earlier in scrapie-infected animals before brain pathology is observed. Of the five body areas examined, the ear pinna and back skin were the areas that showed earliest prion-seeding activity (3 wpi), while the thigh skin was the latest (9 wpi). The latter was also confirmed by two sets of hamsters examined in two-independent laboratories in this study. In contrast to the prion-seeding activity found in the skin of infected hamsters at the early stage of infection, the earliest time point showing skin prion-seeding activity was at 20 wpi in humanized Tg mice by RT-QuIC (Fig. 8b). This time point was similar whether recombinant hamster or human PrP substrate was used despite our expectation that human PrP might provide a more sensitive RT-QuIC for human PrP$^{Sc}$ based on better sequence homology between the seeds and substrate.

Although the reasons for early and widespread presence of PrP$^{Sc}$ in the skin remain unclear, possibilities include the spread of the prion inoculum itself, or endogenously replicating prions, from the brain through the peripheral nerves to the skin within the 2–3 weeks required for the first detection by our ultrasensitive

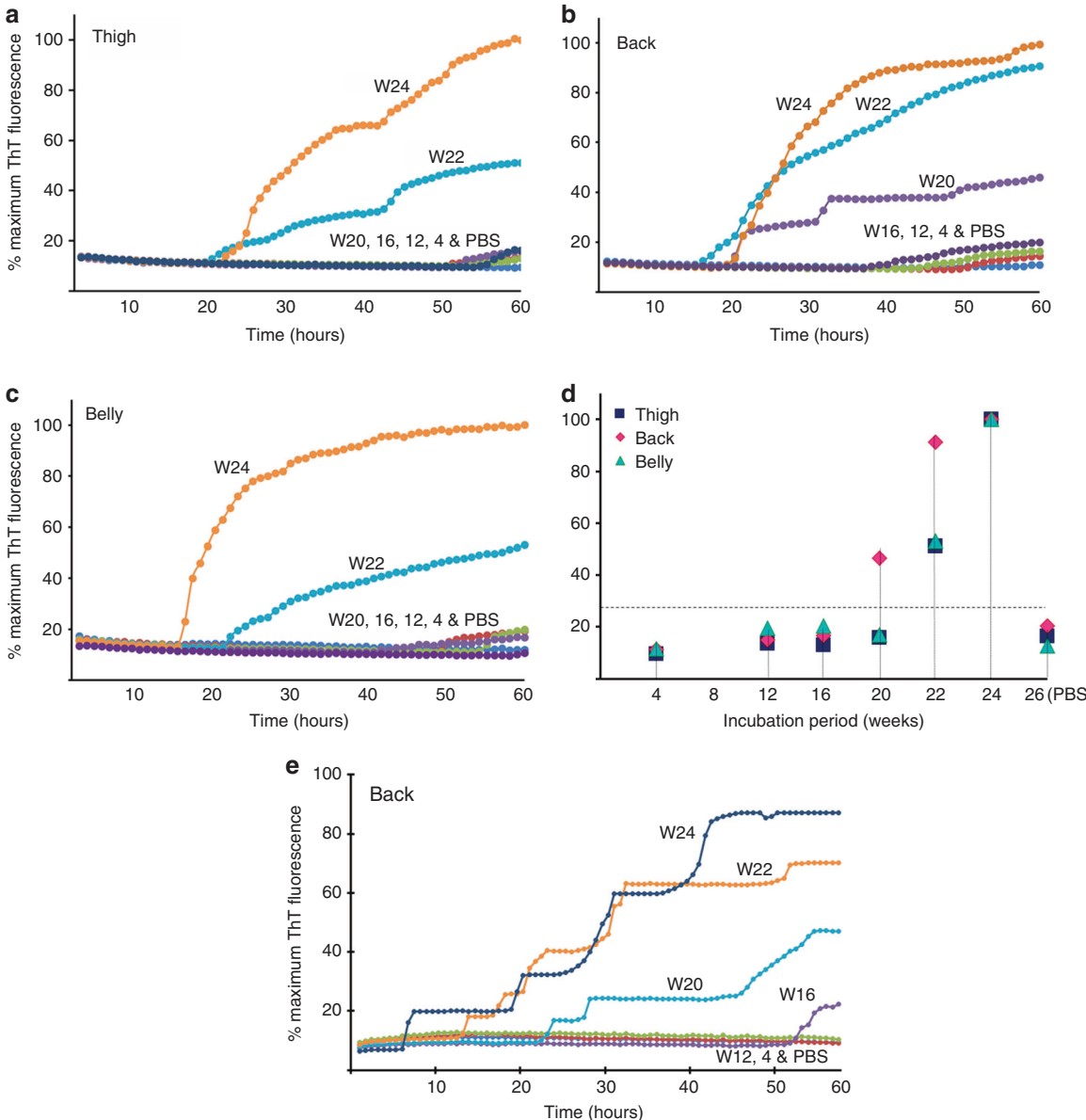

**Fig. 5** Detection of skin prion-seeding activity by RT-QuIC in Tg40h mice. Prion-seeding activity of thigh (**a**) back (**b**) and belly (**c**) skin homogenates from the Tg40h mice inoculated with sCJDMM1 brain homogenate and killed at 4, 8, 12, 16, 20, 22, or 24 wpi was detected by RT-QuIC assay with recombinant hamster PrP90-231 as the substrate. PBS represents-negative control Tg40h mice that were inoculated intracerebrally with PBS for 26 weeks. **d** Scatter plot of average of skin prion-seeding activity of thigh, back, or belly from sCJDMM1-inoculated Tg40h mice at 4, 12, 16, 20, 22, and 24 wpi and PBS-inoculated hamsters killed at 26 wpi. Three animals were examined for each time point. The horizontal dotted line indicates the 27% calculated ThT fluorescence threshold based on the means of negative controls plus 4 standard deviations. **e** Skin samples from back area of sCJDMM1-inoculated Tg40h mice at 4, 12, 16, 20, 22, and 24 wpi and PBS-inoculated control mice were subjected to RT-QuIC assay with recombinant human PrP90-231 as the substrate. The RT-QuIC trace showed is representative of three-independent experiments with three different animals at each time point

sPMCA and RT-QuIC assays. PrP seeding activity has been detected in the blood in the prion-infected hamsters and deer immediately after peripheral inoculation including oral, nasal, or blood route[30]. However, no reports have shown that PrPSc is consistently detectable in the blood of prion-infected hamsters within 2 weeks post intracerebral inoculation. Thus, the early spread of PrPSc from the brain-to-the skin in the intracerebrally 263K-inoculated hamsters is likely either not through the blood or, if initially from the blood, requires time-dependent concentration or replication in the skin to become detectable.

It is unclear why, according to RT-QuIC, the back skin more consistently accumulates PrPSc than the other skin areas tested. It may depend on the dermatomes of nerves and their distance from

the CNS. Between the back and thigh areas examined, the back dermatome is more proximate to the CNS. Similarly, we found prion-seeding activity much earlier in the ear area than the thigh (3 wpi vs 9 wpi). Analogously, misfolded α-synuclein deposition in Parkinson's disease patients is more frequently detected in proximate (100% in the cervical C7 site) compared to distal (35% in the thoracic T12 region) skin areas by immunofluorescence microscopy[31–33]. In future studies, it would be interesting to determine whether PrPSc in the skin of sCJD has a similar distribution, and whether factors besides dermatome distance from the brain are involved.

Both sPMCA and RT-QuIC assays detected skin PrPSc early in scrapie-infected hamsters. However, sPMCA amplified PrPSc in

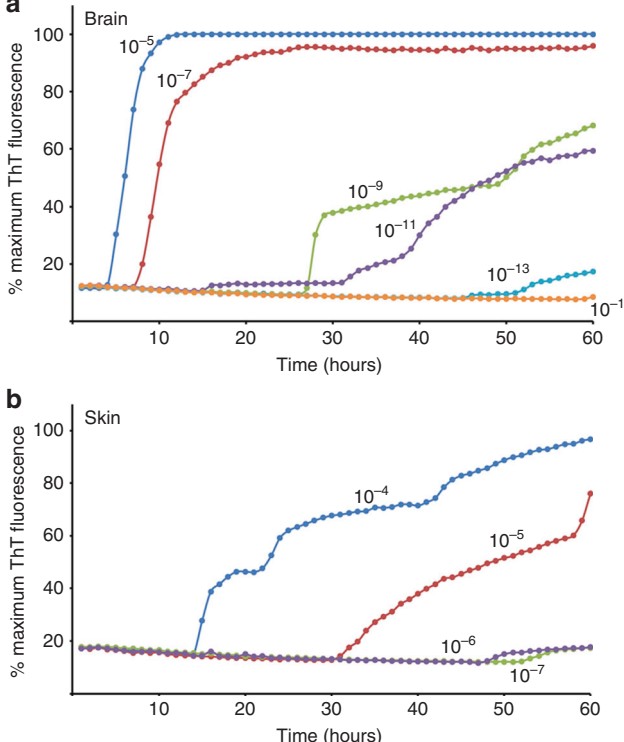

**Fig. 6** RT-QuIC end-point titration of brain and skin samples of Tg40h mice. **a** The brain homogenate from the sCJDMM1-inoculated Tg40h mice was diluted serially until $10^{-15}$ and seeded for RT-QuIC assay. **b** The skin homogenate from the sCJDMM1-inoculated Tg40h mice was serially diluted until $10^{-7}$ and seeded for RT-QuIC assay. Similar results were observed in two-independent experiments. Fluorescence readings from replicate wells were averaged (y-axis) and plotted as a function of time (x-axis). The RT-QuIC spectra are the representative of three-independent experiments with three animals

skin samples from CJD-infected Tg40 mice at 4 wpi, while RT-QuIC assay detected prion-seeding activity only at 20 wpi (Fig. 8). The reason for the difference in Tg40 mice is not clear, but may be due in part to differences between the assays and the prion strains involved. sPMCA is performed in brain homogenates, which provide naturally post-translationally modified (glycosylated and GPI-anchored) $PrP^C$ as the substrate, and other potential brain-derived co-factors. RT-QuIC reactions include only unmodified recombinant $PrP^C$ as substrate, and no natural co-factors. sPMCA reactions are accelerated by sonication, whereas RT-QuIC reactions are shaken. Also, in successive rounds of sPMCA, the substrate and other brain components are refreshed, but our RT-QuIC reactions were performed in one round, with no refreshment. To exclude the effect of mismatch between seeds and substrates on the sensitivity of RT-QuIC reactions, we tested two recombinant PrP molecules as substrates from two different species including hamster and human and they all showed the similar sensitivity with the same earliest time point at 20 wpi. Finally, 263K scrapie and MM1 sCJD prions undoubtedly differ in conformation, and therefore, perhaps, their interactions with co-factors, various $PrP^C$ substrates, and/or skin-derived inhibitors of RT-QuIC reactions. These factors might differentially affect the sensitivity of detection of MM1 sCJD in the skin of Tg40 mice by sPMCA and RT-QuIC. It is also possible that the RT-QuIC assay may become as sensitive as sPMCA for skin prion detection in the Tg40h mice after further optimization of RT-QuIC's experimental conditions.

Our early detection of $PrP^{Sc}$ in the skin of sCJD- and scrapie-infected rodents suggests that it may be possible to do the same with the skin of humans who carry PrP mutations associated with genetic prion diseases such as familial CJD, Gerstmann–Sträussler–Scheinker syndrome, or fatal familial insomnia because it is expected that their mutant $PrP^C$ spontaneously converts into $PrP^{Sc}$ and accumulates later in life. Skin-based RT-QuIC may reveal early prion-seeding activity in PrP mutation-carriers, or people with suspected exposures to prion infections, while they are still asymptomatic. Even for suspected sCJD cases, who are only identified in the symptomatic phase, skin-based RT-QuIC might be useful for monitoring disease progression, defining severity and diversity, and evaluating the treatment efficacy when potential drugs become available.

Although neither clinical signs nor brain $PrP^{Sc}$ were observed in control animals cohabitating with 263K-inoculated hamsters within 12 weeks, the mock-inoculated animals that were housed with scrapie-infected animals had amplifiable $PrP^{Sc}$ in the brain and skin via amplification techniques. Moreover, in contrast to the skin $PrP^{Sc}$ amplified from the 263K-inoculated hamsters as early as 2 wpi, the control animals that co-habitated with infected hamsters were found to have amplified skin $PrP^{Sc}$ after cohabitation for 11 weeks. This finding implies that the skin $PrP^{Sc}$ detected early in the scrapie-infected hamsters is not the result of environmental contamination; otherwise, the control animals would exhibit skin $PrP^{Sc}$ at 2 wpi as well. The finding of skin $PrP^{Sc}$ in the cohabitating control animals may be relevant to the environmental transmission of prions observed in natural animal prion diseases, such as scrapie and CWD. Interestingly, prion transmission has been observed in hamsters by contact with prion-contaminated surfaces through rubbing and bedding[34], in which cases skin is expected to be involved. The role that skin may plays in the environmental transmission of prions warrants future investigation.

Skin $PrP^{Sc}$ may derive from urine or fecal prion contamination in addition to possible skin shedding due to scratching or biting each other. Indeed, scrapie infectivity was reported in the urine of prion-infected mice coincident with lymphocytic nephritis during their preclinical and clinical stages of prion infection[35,36]. It was also observed in their urine in intracerebrally inoculated hamsters even without any apparent inflammation[21]. In addition, deer with clinical CWD and mild to moderate nephritis were found to have sPMCA-detectable $PrP^{Sc}$ and CWD-infectivity in urine[22]. Using sPMCA, $PrP^{Sc}$ was detected in urine of ~80% of the hamsters intraperitoneally inoculated with 263K prions at the symptomatic stage[23]. Notably, $PrP^{Sc}$ was detected in urine, but only at the terminal stage of disease in intracerebrally inoculated hamsters, except for a few days immediately after oral administration[24]. Similar to the observations by Gonzalez-Romero et al.[23], Murayama et al. also found that not all infected hamsters had detectable urine $PrP^{Sc}$ even at the terminal stage[24]. The skin $PrP^{Sc}$ detected early in the intracerebrally infected hamsters, but not in the co-habitated-negative controls, at 2 wpi suggests that skin prions may not result from urine at the early stage of infection.

Unlike the situation with urine, it has not been very clear whether $PrP^{Sc}$ is present in feces of intracerebrally inoculated hamsters at the early stage of prion infection. High titers of prion infectivity were detected in feces throughout the disease incubation in orally inoculated hamsters while low levels of infectivity were occasionally observed in intracerebrally- or intraperitoneally-inoculated animals[18]. For instance, no prion infectivity was detected in feces of hamsters within 3 wpi, including at 1, 2, and 22 days post inoculation (dpi), except for 8 dpi when 17% transmission rate was detected in feces[18]. However, fecal $PrP^{Sc}$ was only detected during the clinical stage of disease by sPMCA in hamsters with lower doses of oral inoculum[19]. Western blotting of fecal extracts showed

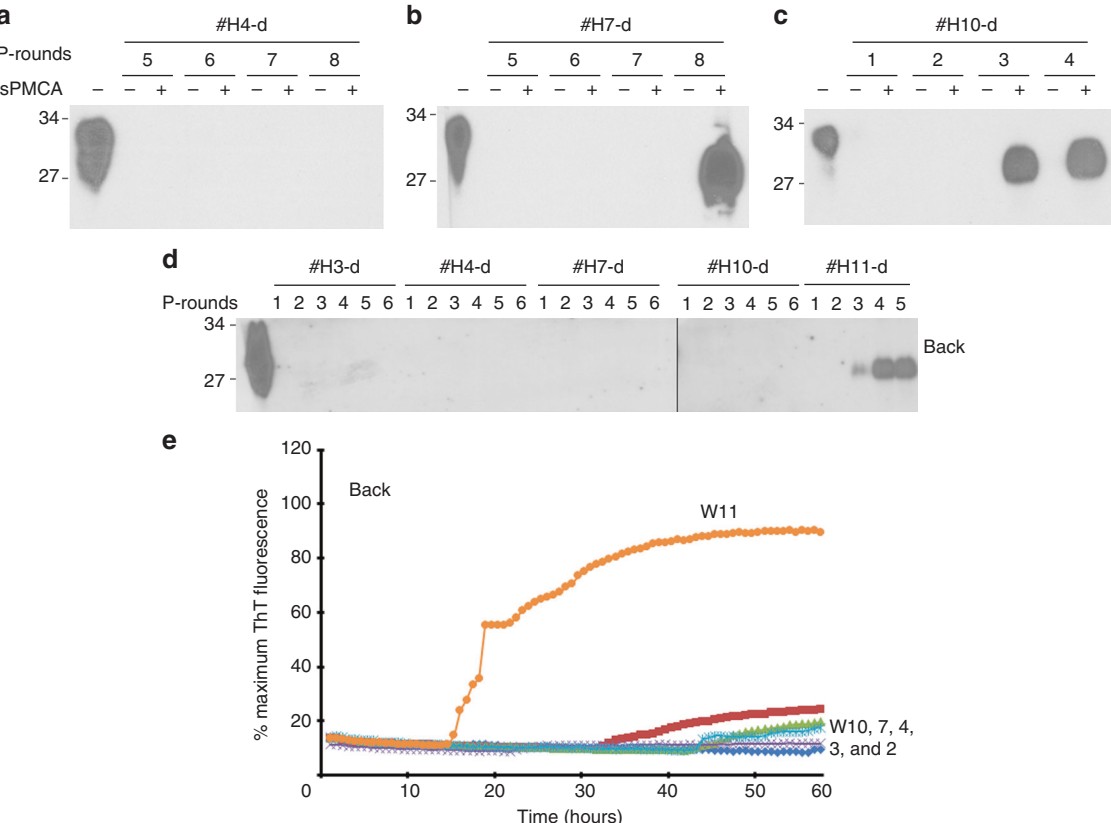

**Fig. 7** PrP$^{Sc}$ in PBS-inoculated hamsters co-habitated with 263K-inoculated hamsters. Representative western blotting of PrP$^{Sc}$ amplified by sPMCA with brain homogenates from PBS-inoculated hamsters in cohabitation with 263K-inoculated hamsters killed at 4 **a**, 7 **b**, 10 **c** wpi from three-independent experiments. **d** Representative western blot analysis of PrP$^{Sc}$ amplified by six rounds of sPMCA from back skin samples from the PBS-inoculated with PBS killed after 3, 4, 7, 10, and 11 weeks of cohabitation with 263K-inoculated hamsters from three-independent experiments. P-rounds: number of sPMCA rounds. All samples were treated with PK at 100 μg/ml except for the sample in the first lane of each blot that was used as a non-PK treatment control. The molecular weight markers are shown in kDa on the left side of each blot. All blots were probed with the 3F4 antibody. **e** Representative of RT-QuIC trace of back skin samples from PBS-inoculated hamsters co-habitated with the 263K-inoculated mice at different time points

shedding of PrP$^{Sc}$ in the excrement at 24–72 h post inoculation, but not at 0–24 h post inoculation, or at later preclinical or clinical time points[19]. Consistent with this observation, prion infectivity was not detected in feces of mule deer after oral challenge with CWD prions within the first 12–16 wpi, but feces contained infectivity after 36 wpi through to clinical disease stages at 64–80 wpi[20]. It is likely that PrP$^{Sc}$ is present in feces of infected animals at the late stage of prion infection, which may contaminate the skin of cohabitating control animals.

Although prion contamination of skin by excrement may not be a major concern in human prion diseases, it is an important issue for prion transmission in animals, such as cattle, sheep, goats, and cervids. It is worth noting that the high incidence of scrapie in sheep and goats as well as CWD in cervids is believed to be attributable to contamination of the environment due to high prion shedding. The detection of PrP$^{Sc}$ in excretions including saliva, urine, and feces clearly indicates this shedding. Oral ingestion due to the coprophagic behavior of animals has been believed to cause wide horizontal transmission of scrapie and CWD. However, our current finding of skin PrP$^{Sc}$ in cohabitating prion-inoculated and PBS-inoculated control animals, as well as the occurrence of brain PrP$^{Sc}$ in the PBS-inoculated animals at the late stage suggests that prion contamination of skin may be a potential route of transmission of prion diseases.

In conclusion, our study indicates that skin PrP$^{Sc}$ may be a useful biomarker not only for the preclinical diagnosis of prion diseases, but also for monitoring disease progression following infection and

treatment. Since the chance that PrP$^{Sc}$ can be consistently detected in blood and urine of sCJD patients by sPMCA and RT-QuIC assays has been virtually very low[10,11,37], it is possible that detection of PrP$^{Sc}$ in the skin, a highly accessible tissue, could be developed for evaluating therapeutic efficiency and drug screening. As mentioned earlier, RT-QuIC analysis of CSF and nasal brushing specimens to date has been used for diagnosis of human prion diseases only at the clinical stage. Moreover, it is much less practical in live animals to collect CSF and nasal brushing specimens. In cervids, at least, there has been more focus on RT-QuIC analyses of RAMALT biopsies and various excreta[38]. Although these analyses are currently the most accurate tests available for chronic wasting disease in live cervids, they do not yet provide 100% diagnostic sensitivity and specificity[38]. Thus, it may be helpful to have additional or alternative diagnostic specimens, such as skin or ear pinna punches, for RT-QuIC and sPMCA testing.

## Methods

**Reagents and antibodies.** Proteinase K (PK) was purchased from Sigma Chemical Co. (St. Louis, MO). Reagents for enhanced chemiluminescence (ECL Plus) were from Amersham Pharmacia Biotech, Inc. (Piscataway, NJ). Anti-PrP mouse monoclonal antibody 3F4 (MAB1562, CHEMICON International, Inc, Burlington, MD, USA) against human PrP residues 107–112[39,40] and sheep anti-mouse IgG conjugated with horseradish peroxidase as a secondary antibody (AC111P, CHEMICON International, Inc, Burlington, MD, USA) were used.

**Inoculation of hamster and Tg40h mice.** Animal inoculation was performed as described below[13,17]. The inoculum was prepared from 263K-infected hamster

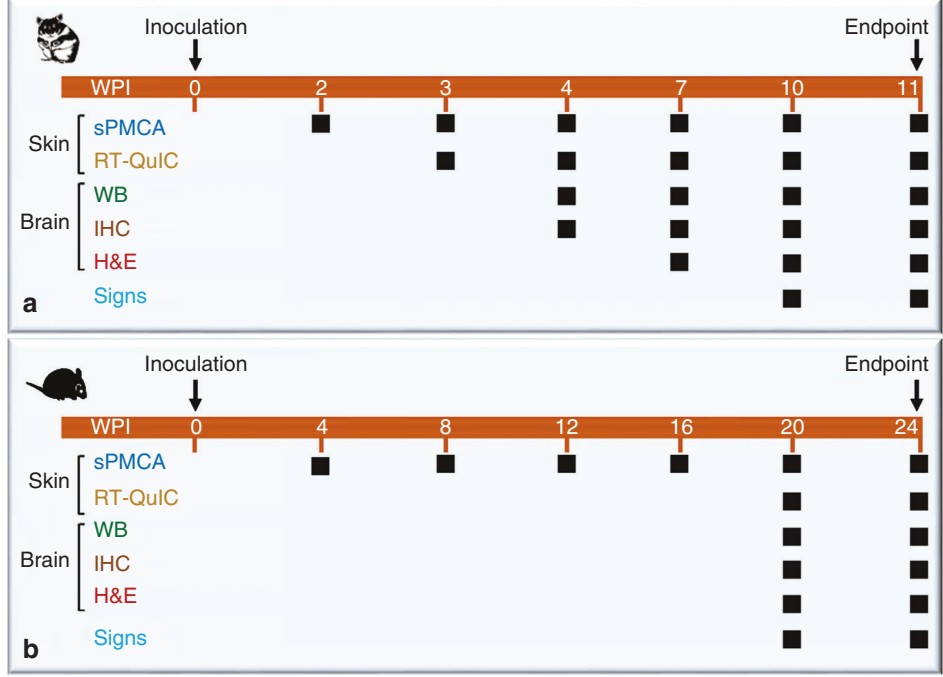

**Fig. 8** Schematic diagram of time-points with detectable PrP$^{Sc}$, brain pathology, or signs. **a** 263K-inoculated hamsters were assessed at different weeks post inoculation (wpi) from day 0 to endpoint at 11 wpi. **b** sCJDMM1-inoculated Tg40h mice were assessed at different wpi from day 0 to endpoint at 24 wpi

brains at terminal stage of infection. A total of 32 female Syrian hamsters at 4 weeks of age were intracerebrally inoculated with 50 μl aliquots of 1% (w/v) 263K-infected hamster brain homogenates in 1X phosphate buffered saline (PBS). The control hamsters were inoculated with 1X PBS and hosted either in a cage with three animals separated from infected animals or co-habitated with three other infected animals in the same cage to determine the possible prion contamination of skin. Another set of hamsters including 52 for intracerebral inoculation with 263K prion strain and 32 for un-inoculation or mock-inoculation controls were conducted for RT-QuIC analysis of the skin and brain samples at the Rocky Mountain NIH laboratory in Hamilton, MT.

A total of 32 Tg40h mice were divided into 8 groups. Each group was housed in one cage and contained four animals, three of which were inoculated with 30 μl of 1% brain homogenate from a patient with sCJDMM1 while one was inoculated with 30 μl of PBS and housed in a separate cage as additional negative control. One group of 4 Tg40h mice was inoculated with 30 μl of PBS as negative controls as well. The inoculated animals were monitored three times a week and were killed at designated time points. The brain and skin tissues were removed and kept frozen at −80 °C for PrP$^{Sc}$ biochemistry test or fixed in 10% PBS-buffered formalin for histological study. Skin samples were taken from different body areas such as ear pinna, back, belly, rib, and thigh for hamsters and back, belly, and thigh for Tg40h mice.

To exclude the possibility of cross-contamination between infected and control animals during the inoculation procedure, the mock inoculations of negative control animals with PBS were conducted first. After the PBS-inoculated animals were placed in their cages, inoculation of animals with the prion-infected brain homogenates were performed. To avoid cross-contamination during the collection of skin samples, the negative control animals were always operated before the prion-infected animals. Moreover, the skin tissues were always sampled always before opening the skull to collect the brain in order to prevent brain-to-skin contamination.

All animal experiments in this study complied with all relevant ethical regulations and were approved by the Institutional Animal Use and Care Committee or the Institutional Biosafety Committee of Case Western Reserve University, or the RML Animal Care and Use Committee.

**Preparation of hamster or mouse skin samples**. After euthanization, the fur was removed by shaving from the areas where skin specimens were taken. Skin samples at 0.5 × 0.5 cm from ear pinna, thigh, back, belly, or rib were collected from hamsters inoculated with 263K scrapie, Tg40h mice, or PBS-negative controls, respectively. The skin specimens consisted of epidermis, dermis, and hypodermis. All skin samples were cut into ~50 mg and stored at −80 °C. The samples were carefully cleaned in Tris-buffered saline (TBS) containing 10 mM Tris-HCl, 133 mM NaCl, pH 7.4 for three times before processing of each specimen in order to avoid blood contamination. For western blotting, or sPMCA assay, the skin samples were incubated in TBS containing 2 mM CaCl₂ and 0.25% (10% w/v) collagenase A (Roche) in a shaker at 37 °C for 4 h. Beads Beater was used to make

tissue homogenates and the samples were then centrifuged at 500×g for 5 min. The supernatant (S1) was transferred to a clean tube for future use and discard the pellet (P1). For RT-QuIC analysis, the S1 fraction was diluted at 1:1 with 2X conversion buffer containing 300 mM NaCl, 2% Triton X-100 and a complete protease inhibitor in PBS without Ca²⁺ and Mg²⁺ to prepare a 5% skin homogenate and then make serial dilution with 0.1% SDS/PBS.

**Serial PMCA procedures**. The preparation of PrP seeds and substrate as well as sPMCA were conducted as follows[14,15]. Hamster or Tg40h mouse brain tissues were carefully dissected to avoid cerebellum and blood contamination as much as possible. Brain homogenate substrates from normal frozen brains were homogenized (10% w/v) in PMCA conversion buffer containing 150 mM NaCl, 1% Triton X-100, 8 mM EDTA pH 7.4 and the complete protease inhibitor mixture cocktail (Roche) in PBS. The seeds of the skin or brain tissue homogenates were prepared as the brain substrates described above. Tissue homogenates were centrifuged at 500×g for 10 min at 4 °C and the supernatant (S1) fraction was collected as the substrate or centrifuged at 500×g for 3 min for the seeds of skin samples. The substrates and seeds were kept at −80 °C until use. Each seed was diluted in the substrate at the ratios 1:12.5 (skin-sPMCA) (8 μl seed + 92 μl substrate) and 1:100 (brain-sPMCA) (1 μl seed + 99 μl substrate) into 200 μl PCR tubes with 1 PTFE beads (diameter 3/32") (Teflon, APT, RI, USA). A 20 μl of each mixture was taken and kept at −20 °C as a non-PMCA control. The remaining mixture was subjected to serial PMCA (sPMCA). Each cycle comprised a 20 s elapse time of sonication at amplitude 85 (250 watts; Misonix S3000 sonicator) followed by an incubation period of 29 min 30 s at 37° and each round of sPMCA consisted of 96 cycles. For the serial PMCA, 20 μl sample was taken from the last cycle and placed into 80 μl fresh normal brain substrates for a new round of amplification.

**Expression and purification of RT-QuIC substrates**. Recombinant truncated Syrian hamster PrP (SHaPrP90-231) and human PrP (HuPrP90–231, with M at residue 129) were prepared as follows[9,41]. *Escherichia coli* carrying a pET41 vector (EMD, Billerica, MA, USA) with the PrP molecule was grown in Luria Broth media containing Kanamycin (0.05 mg/mL) and Chloramphenicol (0.034 mg/mL). Protein expression was induced using Overnight Express Autoinduction System 1 (Novagen, Madison, WI, USA). Inclusion body preparations were isolated from pelleted cells using the Bug Buster Master Mix (Novagen) protocol. An AKTA Prime Plus system (GE Life Sciences, Marlborough, MA) was used for the recombinant protein purification with the addition of a column prewashing step. A denaturing buffer [100 mM sodium phosphate (pH 8.0), 10 mM Tris, 6 M Guanidine-HCl] was passed through the column at 2.3 mL/min for 30 min prior to protein gradient refolding. Protein concentration was determined using a Nano-Drop Lite Spectrophotometer (Thermo Scientific, Wilmington, DE, USA) with absorbance measured at 280 nm. Purity of SHaPrP90-231 and HuPrP90-231 was ≥95%, as estimated by Coomassie blue R-250 dye staining after SDS-PAGE and by

immunoblotting with the 3F4 antibody. Analytical sensitivity and bioactivity of purified SHaPrP90-231 and HuPrP90-231 were tested via the RT-QuIC assay.

**RT-QuIC analysis of hamster or Tg40 mouse skin**. RT-QuIC reaction mix was composed of 10 mM phosphate buffer (pH 7.4), 130 mM or 300 mM NaCl, 0.1 mg/ml truncated recombinant Syrian golden hamster residues 90–231, or human PrP90-231 (129 M), 10 μM Thioflavin T (ThT), 1 mM ethylenediaminetetraacetic acid tetrasodium salt hydrate (EDTA), and 0.001% or 0.002% SDS. Aliquots of the reaction mix (98 μl) were loaded into each well of a 96-well plate (Nunc) and seeded with 2 μl of hamster or Tg40h skin homogenate spun at 2000 g for 2 min at 4 °C as previously described[13]. The plate was then sealed with a plate sealer film (Nalgene Nunc International) and incubated at 42 °C or 55 °C in a BMG FLUOstar Omega plate reader with cycles of 1 min shaking (0.8 g double orbital) and 1 min rest throughout the indicated incubation time. ThT fluorescence measurements (450 ± 10 nm excitation and 480 ± 10 nm emission; bottom read) were taken every 45 min. Four replicate reactions were seeded with the same dilution of an individual sample. The average fluorescence values per sample were calculated using fluorescence values from all four replicate wells regardless of whether these values crossed the threshold described below. At least 2 of 4 replicate wells must cross this threshold for a sample to be considered positive.

End-point dilution titrations were used to quantitate RT-QuIC prion-seeding activity by determining the sample dilution giving positive reactions in 50% of replicates (normally 2 out of 4 replicates) reactions, i.e., the 50% seeding dose or SD$_{50}$. Back calculations then established the SD$_{50}$ per unit of the original specimen[16]. The following equation was used to calculate the log$_{10}$(SD$_{50}$):[16,42,43]

$$\log_{10}SD_{50} = x_{p=l} + 1/2d - d\sum_{X_{p=1}}^{xmin} p_x,\tag{1}$$

in equation (1), $x = \log_{10}$(dilution), $d = \log_{10}$ (dilution factor), $x_{p=1}$ = argument of minimum when $p_x = 1$, and $p$ = proportion positive. In our experiment, serial tenfold dilutions were used, so $d = 1$. $x_{p=1}$ is the most dilute value for which the proportion positive is 1 (positive number/replicate number = 1).

**Western blotting**. sPMCA-treated brain or skin samples were subjected to treatment with proteinase K (PK) at 100 μg/ml for 70 min at 45 °C with agitation prior to western blotting. Samples were resolved on 15% Tris-HCl Criterion precast gels (Bio-Rad) for SDS-PAGE. The proteins on the gels were transferred to Immobilon-P membrane polyvinylidene fluoride (PVDF, Millipore) for 2 h at 70 V. For probing of PrP, the membranes were incubated for 2 h at room temperature with anti-PrP antibody 3F4 at 1:40,000 dilution, as the primary antibody. Following incubation with horseradish peroxidase-conjugated sheep anti-mouse IgG at 1:4000–1:5000 dilution, the PrP bands were visualized on Kodak film by ECL Plus as described by the manufacturer. PrP protein bands were measured by densitometric analysis and quantified using a UN-SCAN-IT Graph Digitizer software (Silk Scientific, Inc., Orem, Utah).

**Hematoxylin & Eosin staining and immunohistochemistry**. Fixed brain or skin tissues received from animals were processed as described below[13]. Sections at 7 mm in thick from the different brain areas or skin tissues were processed for hematoxylin & eosin staining and PrP immunohistochemistry with the anti-PrP monoclonal antibody (mAb) 3F4 at 1:1500 dilutions, 10% GVM and then processed with DAB detection kit as described by the manufacturer.

**Statistical analysis**. Statistical comparisons of mean relative ThT fluorescence responses in samples from scrapie- and PBS-inoculated animals were performed using unpaired Student's $t$-tests.

**Reporting summary**. Further information on experimental design is available in the Nature Research Reporting Summary linked to this article.

## Data availability
The data that support the finding of this study are available from W.-Q.Z. upon reasonable request. A reporting summary for this Article is available as a Supplementary Information file.

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

## Acknowledgements

We thank Miriam Warren and Diane Kofskey for skillful histologic and immunohisto-chemical preparations. We thank Andrew Hughson (RML) for providing recombinant hamster prion protein. Supported by the CJD Foundation and the National Institutes of Health (NIH) NS062787 and NS087588 to W.-Q.Z., NS062787 and NS109532 to W.-Q.Z., and Q.K., NS088604 to Q.K., the Intramural Research Program of the NIAID, NIH to B.C., the Centers for Disease Control and Prevention Contract UR8/CCU515004 to B.S.A. as well as the National Natural Science Foundation of China (No. 81671186) to L.C.

## Author contributions

W.-Q.Z. and B.C. conceived the study. W.-Q.Z., Q.K., L.C., B.C., Z.W., M.M., G.J.R., B.R., and A.F. designed the study. W.-Q.Z., Z.W., J.Y., P.S., J.D., A.A., P.G., R.B.P., X.D., B.S.A., and L.C. developed, performed, and interpreted protein chemistry analysis of PrP$^{Sc}$ in the skin and brain tissues. W.-Q.Z., Z.W., J.Y., and Y.L. developed, performed, and interpreted the sPMCA analysis of skin and brain samples. M.M., A.F., Z.W., C.D.O., B.C., and W.-Q.Z. developed, performed, and interpreted the RT-QuIC analysis of the skin and brain samples. Q.K., W.-Q.Z., G.J.R., B.R., M.V.C., B.L., K.W., N.R.M. and M.M., designed and performed inoculation, monitoring and collection of skin and brain samples from hamsters and humanized transgenic mice. A.F. and Z.W. prepared the recombinant hamster PrP. W.S. provided recombinant hamster PrP for the RT-QuIC analysis at the beginning of the study. B.X. provided reagents. W.-Q.Z. and Z.W. developed, performed, and interpreted the histology data. W.-Q.Z., B.C., C.D.O., Z.W. and M.M. wrote the first version of the paper. All authors critically reviewed, revised- and approved the final version of the manuscript.

## Additional information

**Competing interests:** The authors declare no competing interests.

