## [Peer Review File · Nature Communications]

Reviewers' comments:

Reviewer #1 (Remarks to the Author):

This is a detailed report of preclinical detection of PrPsc/seeding activity in skin samples of hamsters and humanized tg mice infected intracerebrally with 263K scrapie and sCJDMM1 inocula, respectively. Detection methods were sPMCA and RT-QuIC. The new finding is the very early detection of PrPsc/seeding activity in skin at 2 weeks p.i. in hamsters and 4 wks. p.i. in tg mice by sPMCA, whereas RT-QuIC was somewhat less sensitive (3 and 20 wks. p.i., respectively). However, there was quite some variation between sites of skin samples. It was concluded that detection of prions in skin may serve as preclinical biomarker. A bystander result was the demonstration of delayed horizontal transmission by longstanding co-habitation in shared cages.

Clearly, this is an interesting study performed by a recognized consortium with cutting-edge technology that may well be regarded as proof-of-principle investigation. However, the practical and in particular clinical significance is unclear. A lot of different prion diseases and their models have variably shown involvement of peripheral organs, some of which may be similarly accessible to biopsy, and sometimes including preclinical infection states. By more conventional techniques, skin has been demonstrated previously to harbor prions in humans, small ruminants and experimental models; the WHO thus classified skin as "lower infectivity tissue". Using WB and RT-QuIC, skin of symptomatic sCJD patients showed most recently, by an overlapping consortium to the present one, PrPsc/seeding activity that were infectious to tg mice. The present experimental study was similarly designed to preclinical models and performed by additional use of still more sensitive amplification bioassay. Indeed, the present detection methods have become so sensitive that it is not surprising to shift detectability to earlier infection states on one hand, and to less affected tissues on the other. Anyway, it is a different matter whether the original material detected by amplification would be infectious and able to transmit. Unfortunately, the present study lacks infectivity bioassays that would be an attempt to somewhat inform about a potential risk if the present models would really mirror the situation in humans.

Finally, some wording like "Nevertheless, the possibility cannot be ruled out that scrapie-infected hamsters excrete prions into urine at the terminal stage of infection" should be corrected, as shedding of 263K prions in hamster urine has already been demonstrated and cited in another part of the present manuscript.

Reviewer #2 (Remarks to the Author):

This is an interesting paper with important results. The work is well structured and well presented. Minor concern: is not clear from description of the figures how many animals were used for the experiments- only average data or representative data are shown. It would be import to know if the amplification was achieved in all samples or only in part and how was the individual seeding activity

Reviewer #3 (Remarks to the Author):

The manuscript entitled "early preclinical detection of prions in the skin of prion-infected animals" describes the ability of amplification techniques, sPMCA and RT-QuIC, to detect abnormal PrP (PrPsc and prion-seeding activity respectively) in the skin of laboratory animals experimentally infected with prions. Two models are used, i.e. Syrian hamsters exposed to the adapted scrapie strain 263K, and humanized transgenic mice exposed to MM1 sCJD. In both cases, abnormal PrP is detected during the incubation periods, long times before that PrPsc accumulation or lesions were detected within the brains of the animals, or that clinical signs occurred. The authors claim that such techniques performed on skin biopsies may be a biomarker for preclinical diagnosis of prion

disease. In parallel, they show that uninfected animals can be contaminated after long cohabitation periods with infected animals since PrP^{sc} is detected through PMCA in their brain and skin.

Major comments

The present manuscript opens interesting opportunities for preclinical diagnosis of prion diseases in humans but also in animals, for which less invasive blood tests are not currently available. In a previous paper, some of the authors described similar accumulation of PrP^{sc} within the skin of clinical sCJD and vCJD patients: the present work brings the proof of concept in animal models that PrP^{sc} accumulation in skin can occur early during the incubation phase.

The manuscript is well written and convincing, but the description of time results with two different techniques in two experimental models is somehow confusing. It will be of great help for the readers if the authors may provide a summary timeline scheme or a table describing the onset of detection of the different parameters (PrP, lesions, signs) within the different organs at the different times with the different techniques (sPMCA, RT-QuIC, western blot and histology).

On skin samples, sPMCA provided coherent results in both models, with detection occurring very early during the incubation periods (2 and 4 weeks post inoculation in exposed hamsters and mice respectively). In a context of IC inoculation, the distribution of prions within the peripheral organs is supposed to occur secondary after a long time-lag. Which pathophysiological mechanisms might be hypothesized to explain such early, relative high levels of abnormal PrP with wide distribution within the skin of those infected individuals? According to RT-QuIC (and to a lesser extent to sPMCA), skin samples in the back area is significantly more affected than the other areas that have been tested. What is the hypothesis of the authors about this unexpected difference among areas?

RT-QuIC provided different results from sPMCA, with a similar early detection in exposed hamsters (3 weeks post exposure) but after a long time in exposed mice (20 weeks post exposure). A discussion around the absence of correlation between PMCA and RT-Quic results would be appreciated. Notably:

- Titration of seeding activity (SD50) is mentioned without technical explanation. The authors should detail (or reference) this point and notably indicate the numbers of replicates that allow measurement of SD50. Notably, why is SD50 calculated with precision for hamsters but not for mice ?
- According to RT-QuIC, seeding activities are within the same order of magnitude in the skin samples of both models, but seeding activity in brains of mice (9 to 11 logs SD50/mg) is higher than in the brains of hamsters (8.75 to 9.5 logs SD50/mg). Conversely, brains from 263K-infected hamsters are classically reputed to harbor higher infectious titers than prion-infected mice models. Is this model of transgenic mice peculiar? How can we explain this apparent paradox whereas hamster recombinant PrP (thus supposed to be more adapted to hamsters than to mice) is used as substrate? Why did the authors use hamster recombinant PrP for RT-QuIC with tg40 mice samples, and not human recombinant PrP?

The authors assumed that the presence of PrP^{sc} in the skin of exposed animals is due to the experimental exposure and is not the result of an inter-animal contamination, since it was not found with the same timeframe in the skin of non-inoculated animals that were housed in the same cages as these inoculated animals. Nevertheless, the authors showed that these control animals are also contaminated since they first detected PrP^{sc} with sPMCA techniques in their brain (but not through western blot), and then later in their skin. Non-controlled contamination between animals housed in same cages may thus occur, and the authors proposed in their discussion several hypotheses to explain such events.

However, I think two points are lacking in the hamster model for a better comparison between the two groups (exposed versus control animals in cohabitation):

- Did the authors detect PrP seeding activity in the skin of controlled hamsters in cohabitation through RT-QuIC?
- At which time of the incubation periods of experimentally-infected animals did the authors detect PrPsc with sPMCA?

In parallel, the authors should provide experimental details to rule out the possibility of cross-contamination between exposed animals and control animals during the period surrounding the i.c. inoculation, and a potential cross-contamination of samples at shaving.

Minor comments

Page 7 line 3: a difference of 5 logs is mentioned between titres of skin and brain samples; according to table I, the difference ranges from 3.75 to 5.75: please be more precise on this point.

Page 15 line 14: Teflon and not Telflon

In the authors' contributions, "C.L. designed the study". Who is C.L. in the authors? The role of several authors has not been detailed in the authors' contribution section.

The Material sections is lacking ethical statement.

Please precise the number of animals tested at each time point for statistical significance.

For a better reading of figures 2 and 3, please mention the location of skin samples (back-belly-thigh) on the top of the diagrams a), b) and c). For captions 2d and 3d, please be homogenous on the position of PBS controls (T=0 or at the end of the graph). Please also precise "incubation period" for the X axis (to avoid confusion with "time" of RT-QuIC in captions a b and c).

Fig S1: what are "11a" and "11b" weeks of exposure?

Fig S5: please precise the inoculation performed to animal identified as a, b, c and d (d= PBS inoculated ?)

Point-by-point response to the referees' comments:

Reviewer #1 (Remarks to the Author):

This is a detailed report of preclinical detection of PrP^{Sc}/seeding activity in skin samples of hamsters and humanized tg mice infected intracerebrally with 263K scrapie and sCJDMM1 inocula, respectively. Detection methods were sPMCA and RT-QuIC. The new finding is the very early detection of PrP^{Sc}/seeding activity in skin at 2 weeks p.i. in hamsters and 4 wks. p.i. in tg mice by sPMCA, whereas RT-QuIC was somewhat less sensitive (3 and 20 wks. p.i., respectively). However, there was quite some variation between sites of skin samples. It was concluded that detection of prions in skin may serve as preclinical biomarker. A bystander result was the demonstration of delayed horizontal transmission by longstanding co-habitation in shared cages.

Response: Indeed, variations in prion-seeding activity between different sites of skin samples were observed, which suggests different levels of PrP^{Sc} in skin from these sites, possibly related to the corresponding spinal vertebral levels of their respective dermatome. It is worth noting that, in skin of patients with Parkinson's disease, immunofluorescence microscopy revealed higher positive rate of misfolded α -synuclein deposits in the cervical sites than in the thoracic sites (100% vs 35%) (Donadio et al., 2017). Currently, we are in the process of ascertaining whether skin PrP^{Sc} distribution in sCJD patients exhibit the same pattern using skin-based RT-QuIC assay, which unfortunately will not be included in this study because it will take a lot time to collect skin samples from multiple sites of an adequate number of sCJD subjects.

Clearly, this is an interesting study performed by a recognized consortium with cutting-edge technology that may well be regarded as proof-of-principle investigation. However, the practical and in particular clinical significance is unclear.

Response: We thank the referee for this important question. As noted by Reviewer #3, "*The present manuscript opens interesting opportunities for preclinical diagnosis of prion diseases in humans but also in animals, for which less invasive blood tests are not currently available*". We agree, and have added the following discussion in "Discussion" section. "Our early detection of PrP^{Sc} in the skin of sCJD- and scrapie-infected rodents suggests that it may be possible to do the same with the skin of humans who carry PrP mutations associated with genetic prion diseases such as familial CJD, Gerstmann-Sträussler-Scheinker syndrome, or fatal familial insomnia because it is expected that their mutant PrP^C spontaneously converts into PrP^{Sc} and accumulates later in life. Skin-based RT-QuIC may reveal early prion-seeding activity in PrP mutation-carriers, or people with suspected exposures to prion infections, while they are still asymptomatic. Even for suspected sCJD cases, who are only identified in the symptomatic phase, skin-based RT-QuIC might be useful for monitoring disease progression, defining severity and diversity, and evaluating the treatment efficacy when potential drugs become available."

A lot of different prion diseases and their models have variably shown involvement of peripheral organs, some of which may be similarly accessible to biopsy, and sometimes including preclinical infection states. By more conventional techniques, skin has been demonstrated previously to harbor prions in humans, small ruminants and experimental models; the WHO thus classified skin as "lower infectivity tissue". Using WB and RT-QuIC, skin of symptomatic sCJD patients showed most recently, by an overlapping consortium to the present one, PrP^{Sc}/seeding activity that were infectious to tg mice. The present experimental study was similarly designed to preclinical models and performed by additional use of still more sensitive amplification bioassay. Indeed, the present detection methods have become so sensitive that it is not surprising to shift detectability to earlier infection states on one hand, and to less affected

tissues on the other. Anyway, it is a different matter whether the original material detected by amplification would be infectious and able to transmit. Unfortunately, the present study lacks infectivity bioassays that would be an attempt to somewhat inform about a potential risk if the present models would really mirror the situation in humans.

Response: We appreciate that the referee raises this issue. We agree with the referee that it would be interesting to determine whether skin samples from the early stage of the prion-infected animals are infectious, as observed with the skin homogenate from sCJD patients. We are unable to include this animal-based transmission study in this manuscript because it will take more than 4 months for such a study given the expected low infectivity compared to the brain tissues. However, given that we and others have already shown the presence of infectious prions in the skin of prion-infected individuals, our primary goal here is to see how early we can detect signs of infection using an assay of potential diagnostic utility (which is not the case for animal bioassays).

Finally, some wording like “Nevertheless, the possibility cannot be ruled out that scrapie-infected hamsters excrete prions into urine at the terminal stage of infection” should be corrected, as shedding of 263K prions in hamster urine has already been demonstrated and cited in another part of the present manuscript.

Response: It is true that shedding of 263K prions in hamster urine has been reported. We changed the sentence as “In fact, scrapie-infected hamsters have been reported to excrete prions into urine at the terminal stage of infection”.

Reviewer #2 (Remarks to the Author):

This is an interesting paper with important results. The work is well structured and well presented. Minor concern: is not clear from description of the figures how many animals were used for the experiments- only average data or representative data are shown. It would be import to know if the amplification was achieved in all samples or only in part and how was the individual seeding activity

Response: We wanted to thank the Reviewer for the positive comments and for raising the question about numbers. Now we include the information about number of animals used in each experiment described in figure legends.

Reviewer #3 (Remarks to the Author):

The manuscript entitled “early preclinical detection of prions in the skin of prion-infected animals” describes the ability of amplification techniques, sPMCA and RT-QuIC, to detect abnormal PrP (PrPsc and prion-seeding activity respectively) in the skin of laboratory animals experimentally infected with prions. Two models are used, i.e. Syrian hamsters exposed to the adapted scrapie strain 263K, and humanized transgenic mice exposed to MM1 sCJD. In both cases, abnormal PrP is detected during the incubation periods, long times before that PrPsc accumulation or lesions were detected within the brains of the animals, or that clinical signs occurred. The authors claim that such techniques performed on skin biopsies may be a biomarker for preclinical diagnosis of prion disease. In parallel, they show that uninfected animals can be contaminated after long cohabitation periods with infected animals since PrPsc is detected through PMCA in their brain and skin.

Major comments

The present manuscript opens interesting opportunities for preclinical diagnosis of prion diseases in humans but also in animals, for which less invasive blood tests are not currently available. In a previous paper, some of the authors described similar accumulation of PrPsc within the skin of clinical sCJD and

vCJD patients: the present work brings the proof of concept in animal models that PrP^{Sc} accumulation in skin can occur early during the incubation phase.

The manuscript is well written and convincing, but the description of time results with two different techniques in two experimental models is somehow confusing. It will be of great help for the readers if the authors may provide a summary timeline scheme or a table describing the onset of detection of the different parameters (PrP, lesions, signs) within the different organs at the different times with the different techniques (sPMCA, RT-QuIC, western blot and histology).

Response: Thank the Referee for the excellent suggestion. A diagram is added as Fig. 5 to show assessment time points from inoculation to death of animals in the revised manuscript.

On skin samples, sPMCA provided coherent results in both models, with detection occurring very early during the incubation periods (2 and 4 weeks post inoculation in exposed hamsters and mice respectively). In a context of IC inoculation, the distribution of prions within the peripheral organs is supposed to occur secondary after a long time-lag. Which pathophysiological mechanisms might be hypothesized to explain such early, relative high levels of abnormal PrP with wide distribution within the skin of those infected individuals?

Response: We thank the Referee for this interesting question. We added the following text in the “Discussion” section of the revised manuscript. “Although the reasons for early and widespread presence of PrP^{Sc} in the skin remain unclear, possibilities include the spread of the prion inoculum itself, or endogenously replicating prions, from the brain through the peripheral nerves to the skin within the 2-3 weeks required for the first detection by our ultrasensitive sPMCA and RT-QuIC assays. PrP seeding activity has been detected in the blood in the prion-infected hamsters and deer immediately after peripheral inoculation including oral, nasal, or blood route (Elder et al., 2015). However, no reports have shown that PrP^{Sc} is consistently detectable in the blood of prion-infected hamsters within two weeks post intracerebral inoculation. Thus, the early spread of PrP^{Sc} from the brain to the skin in the intracerebrally 263K-inoculated hamsters is likely either not through the blood or, if initially from the blood, requires time-dependent concentration or replication in the skin to become detectable.”

According to RT-QuIC (and to a lesser extent to sPMCA), skin samples in the back area is significantly more affected than the other areas that have been tested. What is the hypothesis of the authors about this unexpected difference among areas?

Response: We added the following text to the “Discussion” section: “It is unclear why, according to RT-QuIC, the back skin more consistently accumulates PrP^{Sc} than the other skin areas tested. It may depend on the dermatomes of nerves and their distance from the CNS. Between the back and thigh areas examined, the back dermatome is more proximate to the CNS. Similarly, we found prion-seeding activity much earlier in the ear area than the thigh (3 wpi vs 9 wpi). Analogously, misfolded α -synuclein deposition in Parkinson’s disease patients is more frequently detected in proximate (100% in the cervical C7 site) compared to distal (35% in the thoracic Th12 region) skin areas by immunofluorescence microscopy (Dondio et al., 2017; Doppler et al., 2014; Donadio et al., 2014). In future studies, it would be interesting to determine whether PrP^{Sc} in the skin of sCJD has a similar distribution, and whether factors besides dermatome distance from the brain are involved.”

RT-QuIC provided different results from sPMCA, with a similar early detection in exposed hamsters (3 weeks post exposure) but after a long time in exposed mice (20 weeks post exposure). A discussion around the absence of correlation between PMCA and RT-Quic results would be appreciated.

Response: We thank the Referee for raising this issue. Following the Referee’s suggestion, we added the following discussion in the revised manuscript: “Both sPMCA and RT-QuIC assays detected skin PrP^{Sc}

early in scrapie-infected hamsters. However, sPMCA amplified PrP^{Sc} in the skin samples from CJD-infected Tg40 mice at 4 wpi while RT-QuIC assay detected prion-seeding activity only at 20 wpi (Fig. 5). The reason for the difference in Tg40 mice is not clear, but may be due in part to the differences between the assays and the prion strains involved. sPMCA is performed in brain homogenates, which provide naturally post-translationally modified (glycosylated and GPI-anchored) PrP^C as the substrate, and other potential brain-derived co-factors. RT-QuIC reactions include only unmodified recombinant PrP^C as substrate, and no natural cofactors. sPMCA reactions are accelerated by sonication, whereas RT-QuIC reactions are shaken. Also, in successive rounds of sPMCA, the substrate and other brain components are refreshed, but our RT-QuIC reactions were performed in one round, with no refreshment. To exclude the effect of mismatch between seeds and substrates on the sensitivity of RT-QuIC reactions, we tested two recombinant PrP molecules as substrates from two different species including hamster and human and they all showed the similar sensitivity with the same earliest time point at 20 wpi. Finally, 263K scrapie and MM1 sCJD prions undoubtedly differ in conformation, and therefore, perhaps, their interactions with cofactors, various PrP^C substrates, and/or skin-derived inhibitors of RT-QuIC reactions. These factors might differentially affect the sensitivity of detection of MM1 sCJD in the skin of Tg40 mice by sPMCA and RT-QuIC.”

Notably:

- Titration of seeding activity (SD50) is mentioned without technical explanation. The authors should detail (or reference) this point and notably indicate the numbers of replicates that allow measurement of SD50. Notably, why is SD50 calculated with precision for hamsters but not for mice?

Response: We add SD₅₀ calculation for infected mice in the revised manuscript including technical explanation and the number of replicates under “Materials and Methods” section as follows: “End-point dilution titrations were used to quantitate RT-QuIC prion seeding activity by determining the sample dilution giving positive reactions in 50% of replicates (normally 2 out of 4 replicates) reactions, i.e., the 50% seeding dose or SD₅₀. Back calculations then established the SD₅₀ per unit of the original specimen (Wilham et al., 2010). The following equation was used to calculate the log₁₀(SD₅₀) as previously described (Hamilton et al., 1977; Peden et al., 2012; Wilham et al., 2010):

$$\log_{10}SD_{50} = x_{p=1} + 1/2d - d \sum_{xP=1}^{x_{min}} p_x$$

in which $x = \log_{10}(\text{dilution})$, $d = \log_{10}(\text{dilution factor})$, $x_{p=1} = \text{argmin}_x (p_x = 1)$, and $p = \text{proportion positive}$. In our experiment, serial 10-fold dilutions were used, so $d = 1$. $x_{p=1}$ is the most dilute value for which the proportion positive is 1 (positive number/replicate number = 1).

- According to RT-QuIC, seeding activities are within the same order of magnitude in the skin samples of both models, but seeding activity in brains of mice (9 to 11 logs SD50/mg) is higher than in the brains of hamsters (8.75 to 9.5 logs SD50/mg). Conversely, brains from 263K-infected hamsters are classically reputed to harbor higher infectious titers than prion-infected mice models. Is this model of transgenic mice peculiar? How can we explain this apparent paradox whereas hamster recombinant PrP (thus supposed to be more adapted to hamsters than to mice) is used as substrates? Why did the authors use hamster recombinant PrP for RT-QuIC with tg40 mice samples, and not human recombinant PrP?

Response: The Reviewer is correct that brains from 263K-infected hamsters often have higher infectious titers than prion-infected mice, which is also shown in our results on end-point titration with the brain and skin samples of hamsters and Tg40h mice (see newly-added Table 1), which was all done in the same lab). As shown in the figures, indeed, the seeding activity in the brain of hamsters is approximately 13 logs SD₅₀/mg which is higher than that in Tg40h mouse brain (9-11 logs SD₅₀/mg). The apparent paradox could be due to the two datasets were from two different labs. For instance, the data on 263K-

infected hamsters shown in Tables 1, 2 were generated in two different labs. It is possible that different machines and experiment conditions used in the two labs could result in some variations. Now we include two sets of data as Tables 1, 2 in the revised manuscript. The reason that the recombinant hamster PrP were used as substrates in our study is because our previous studies revealed that both recombinant hamster and bank vole PrP worked well for human PrP^{Sc} (Orru et al., 2011; Orru et al., 2017). To specifically address the Referee's concern, we examined the back skin of Tg40h mice inoculated with sCJDMM1 brain homogenate using human recombinant PrP90-231 as the substrate in RT-QuIC assay. Our result showed that no significant difference in sensitivity between hamster and human PrP was observed, which was included as Fig. S6 in the revised manuscript along with the following description added in "Results" section "To determine whether the use of recombinant human PrP as the substrate could increase sensitivity, we examined the back skin samples from infected Tg40h mice at different time points with the HuPrP(90-231)-based RT-QuIC assay. Similar to results shown in Fig. 3C, prion-seeding activity was only detected in the skin of Tg40h mice at 20 wpi and afterwards (Fig. S6). No seeding activity was observed in the skin of mice at 16 wpi or earlier."

The authors assumed that the presence of PrPsc in the skin of exposed animals is due to the experimental exposure and is not the result of an inter-animal contamination, since it was not found with the same timeframe in the skin of non-inoculated animals that were housed in the same cages as these inoculated animals. Nevertheless, the authors showed that these control animals are also contaminated since they first detected PrPsc with sPMCA techniques in their brain (but not through western blot), and then later in their skin. Non-controlled contamination between animals housed in same cages may thus occur, and the authors proposed in their discussion several hypotheses to explain such events.

However, I think two points are lacking in the hamster model for a better comparison between the two groups (exposed versus control animals in cohabitation):

- Did the authors detect PrP seeding activity in the skin of controlled hamsters in cohabitation through RT-QuIC?

Response: Yes, we detected prion seeding activity in the skin of controlled hamsters in cohabitation with RT-QuIC assay and now we include it in the updated Fig. 4 in the revised manuscript.

- At which time of the incubation periods of experimentally-infected animals did the authors detect PrPsc with sPMCA?

Response: We collected the skin samples right after inoculation of hamsters with 263K-infected hamster brain homogenate at 0, 0.4, 1, 2, ...12 weeks post inoculation. Skin PrP^{Sc} was detectable at 2 wpi and afterwards (Fig. 1) but was not detectable at 0, 0.4 and 1 wpi in the skin of hamsters inoculated intracerebrally by sPMCA-based Western blotting (Fig. S2).

In parallel, the authors should provide experimental details to rule out the possibility of cross-contamination between exposed animals and control animals during the period surrounding the i.c. inoculation, and a potential cross-contamination of samples at shaving.

Response: We added detailed procedures of inoculation of both 263K prion and PBS. We excluded the possibility of cross-contamination during inoculation and at shaving by inoculating with PBS separately from inoculating with infected brain homogenates, as well as separation of shaving with dedicated shavers of control animals from infected animals. We added the following sentence in the section "Inoculation of hamster and Tg40h mice" under "Methods": "To exclude the possibility of cross-contamination between infected and control animals during the inoculation procedure, the mock inoculations of negative control animals with PBS were conducted first. After the PBS-inoculated animals were caged, we performed the inoculations of animals with the prion-infected brain homogenates."

Moreover, the skin tissues were always sampled before opening the skull to collect the brain in order to prevent brain-to-skin contamination.”

Minor comments

Page 7 line 3: a difference of 5 logs is mentioned between titres of skin and brain samples; according to table 1, the difference ranges from 3.75 to 5.75: please be more precise on this point.

Response: We double-checked titers of prions between skin and brain samples and changed the sentence as the follows to make sure that they are precise. “End-point dilution RT-QuIC reactions of 11-12 wpi tissues indicated that the average prion-seeding activity in skin samples was about 10^3 - to 10^5 -fold lower than that in brain tissues (Tables 1, 2).”

Page 15 line 14: Teflon and not Telflon

Response: Thank the Reviewer for the correction. The error is corrected.

In the authors’ contributions, “C.L. designed the study”. Who is C.L. in the authors? The role of several authors has not been detailed in the authors’ contribution section.

Response: Sorry for the errors. It should be “L.C.” referring to Li Cui. The role of several authors missed have now been included in the authors’ contribution section.

The Material sections is lacking ethical statement.

Response: Ethical statement is included in the revised manuscript as follows “The study was monitored and approved by the University Hospitals Case Medical Center Institutional Review Board. All animal experiments in this study were approved by the Institutional Animal Use and Care Committee and the Institutional Biosafety Committee of Case Western Reserve University, or the RML Animal Care and Use Committee (Protocol #2016-039E)..” under “Materials and Methods” section.

Please precise the number of animals tested at each time point for statistical significance.

Response: The number of animals tested at each time point for statistical significance is included in the figure legends of revised manuscript.

For a better reading of figures 2 and 3, please mention the location of skin samples (back-belly-thigh) on the top of the diagrams a), b) and c). For captions 2d and 3d, please be homogenous on the position of PBS controls (T=0 or at the end of the graph). Please also precise “incubation period” for the X axis (to avoid confusion with “time” of RT-QuIC in captions a b and c).

Response: Following the Reviewer’s suggestions, the location of skin samples examined is indicated in Figs. 2 and 3 on the top of each panel. To be consistent, the position of PBS controls is now in the same place on Figs. 2d and 3e. “Incubation period” is used for the X axis in panels 2d and 3e.

Fig S1: what are “11a” and “11b” weeks of exposure?

Response: “11a” and “11b” refer to the hamsters that had the same incubation time but were housed in two different cages. A description of them is added in the figure legend to Fig. S1.

Fig S5: please precise the inoculation performed to animal identified as a, b, c and d (d= PBS inoculated?)

Response: A description for the inoculation on animal ID a, b, c and d is added in the figure legend to Fig. S5.

REVIEWERS' COMMENTS:

Reviewer #1 (Remarks to the Author):

In the revision, authors have acknowledged and addressed most of the reviewers' concerns in an adequate way. The reason for the significant variability between sites of skin samples, however, remains unclear, as is the clinical significance. The authors now added a small section in the discussion to elaborate further on this issue. It is understandable that infectivity bioassay would be lengthy and probably reported in a subsequent study. Useful additions now also include ethical statements, figures on animal numbers, cohabitation experimental results, discussion on differences between both methodologies and both models, SD50 calculations, description of RT-QuIC substrate, exclusion of cross-contamination, improved lettering of figures, and a new figure for clarification of experimental timelines.

Reviewer #2 (Remarks to the Author):

The authors addressed all comments raised by the reviewer well. Please check Figure 5, it seems that the description of different time points is misplaced

Reviewer #3 (Remarks to the Author):

The authors correctly answered the different questions and remarks I had on their initial manuscript, and I thank them for this. Values of SD50 are now more coherent between the animal models, and the discussion addresses most of the questions highlighted by these results, even if no definitive answer is provided for some of them.

Figure 5 is useful to guide the reader, even if a more graphic figure would have been appreciated. In the legend, this figure is mentioned as "a schematic diagram of the assessment time points from inoculation to animal death", whereas it is rather describing the time of onset of detection of the different parameters (PrP, lesions, signs) within the different organs at the different times with the different techniques (sPMCA, RT-QuIC, western blot and histology). I renew my congratulations to the authors for this interesting work which has several consequences in our understanding of prion physiopathology and applications in terms of diagnosis.

Point-by-point response

REVIEWERS' COMMENTS:

Reviewer #1 (Remarks to the Author):

In the revision, authors have acknowledged and addressed most of the reviewers' concerns in an adequate way. The reason for the significant variability between sites of skin samples, however, remains unclear, as is the clinical significance. The authors now added a small section in the discussion to elaborate further on this issue. It is understandable that infectivity bioassay would be lengthy and probably reported in a subsequent study. Useful additions now also include ethical statements, figures on animal numbers, cohabitation experimental results, discussion on differences between both methodologies and both models, SD50 calculations, description of RT-QuIC substrate, exclusion of cross-contamination, improved lettering of figures, and a new figure for clarification of experimental timelines.

Response: We thank the Referee for the positive comments. As the Referee indicated, we included a brief description to discuss the possible reasons about the variability in the levels of PrPSc of different sites of skin samples and its clinical significance. These issues are important and warrant further investigation in the future.

Reviewer #2 (Remarks to the Author):

The authors addressed all comments raised by the reviewer well. Please check Figure 5, it seems that the description of different time points is misplaced.

Response: We want to thank the Referee for the positive comments and for the constructive suggestion of Fig. 5. It has been changed accordingly in the revised manuscript.

Reviewer #3 (Remarks to the Author):

The authors correctly answered the different questions and remarks I had on their initial manuscript, and I thank them for this. Values of SD50 are now more coherent between the animal models, and the discussion addresses most of the questions highlighted by these results, even if no definitive answer is provided for some of them.

Figure 5 is useful to guide the reader, even if a more graphic figure would have been appreciated. In the legend, this figure is mentioned as "a schematic diagram of the assessment time points from inoculation to animal death", whereas it is rather describing the time of onset of detection of the different parameters (PrP, lesions, signs) within the different organs at the different times with the different techniques (sPMCA, RT-QuIC, western blot and histology). I renew my congratulations to the authors for this interesting work which has several consequences in our understanding of prion physiopathology and applications in terms of diagnosis.

Response: We would like to thank the Referee so much for the positive comments and constructive suggestion. We revised the title of Fig. 5 and provided a more graphic figure (now Fig. 8) as follows: "Schematic diagram of time-points with detectable PrPSc, brain pathology or signs."